# Integrating Casein Complex SNPs Additive, Dominance and Epistatic Effects on Genetic Parameters and Breeding Values Estimation for Murciano-Granadina Goat Milk Yield and Components

**DOI:** 10.3390/genes11030309

**Published:** 2020-03-14

**Authors:** María Gabriela Pizarro Inostroza, Vincenzo Landi, Francisco Javier Navas González, Jose Manuel León Jurado, Juan Vicente Delgado Bermejo, Javier Fernández Álvarez, María del Amparo Martínez Martínez

**Affiliations:** 1Department of Genetics, Faculty of Veterinary Sciences, University of Córdoba, 14071 Córdoba, Spain; z12piinm@uco.es (M.G.P.I.); id1debej@uco.es (J.V.D.B.); ib2mamaa@uco.es (M.d.A.M.M.); 2Animal Breeding Consulting, S.L., Córdoba Science and Technology Park Rabanales 21, 14071 Córdoba, Spain; 3Department of Veterinary Medicine, University of Bari “Aldo Moro”, 70010 Valenzano, Italy; vincenzo.landi@uniba.it; 4Centro Agropecuario Provincial de Córdoba, Diputación Provincial de Córdoba, Córdoba, 14071 Córdoba, Spain; jomalejur@yahoo.es; 5National Association of Breeders of Murciano-Granadina Goat Breed, Fuente Vaqueros, 18340 Granada, Spain; j.fernandez@caprigran.com

**Keywords:** OVERALS, CATPCA, Kruskal–Wallis, association, restricted maximum likelihood, linkage disequilibrium, casein complex, MTDFREML

## Abstract

Assessing dominance and additive effects of casein complex single-nucleotide polymorphisms (SNPs) (αS1, αS2, β, and κ casein), and their epistatic relationships may maximize our knowledge on the genetic regulation of profitable traits. Contextually, new genomic selection perspectives may translate this higher efficiency into higher accuracies for milk yield and components’ genetic parameters and breeding values. A total of 2594 lactation records were collected from 159 Murciano-Granadina goats (2005–2018), genotyped for 48 casein loci-located SNPs. Bonferroni-corrected nonparametric tests, categorical principal component analysis (CATPCA), and nonlinear canonical correlations were performed to quantify additive, dominance, and interSNP epistatic effects and evaluate the outcomes of their inclusion in quantitative and qualitative milk production traits’ genetic models (yield, protein, fat, solids, and lactose contents and somatic cells count). Milk yield, lactose, and somatic cell count heritabilities increased considerably when the model including genetic effects was considered (0.46, 0.30, 0.43, respectively). Components standard prediction errors decreased, and accuracies and reliabilities increased when genetic effects were considered. Conclusively, including genetic effects and relationships among these heritable biomarkers may improve model efficiency, genetic parameters, and breeding values for milk yield and composition, optimizing selection practices profitability for components whose technological application may be especially relevant for the cheese-making dairy sector.

## 1. Introduction

The Murciano-Granadina goat is one of the main dairy goat breeds in Spain. This special relevance has been achieved through its census, geographical distribution, and its milk quality and production, which on the whole, is well-suited for the dairy industry [1].

The demands for goat milk have increased in recent decades, hence selection programs have faced the need to adapt through new technological and methodological advances to more suitably and appropriately respond to them. Contextually, the optimization of genetic evaluations has become an important point to consider these days, as the success or failure of any potential selection strategy for a certain trait fundamentally depends on the accuracy with which we are able to evaluate it. 

The production of milk and its components is regulated by complex inheritance mechanisms and is influenced by many pairs of genes whose expression depends on the interaction with the environment in which the animals are located. One of the most relevant sets of genes involved in the regulation of the expression of milk quantity and quality is the casein complex. The genetic variants for caseins (αS1, αS2, β, and κ casein) influence the traits related to the components of milk and its profitability [2]. In this sense, the estimation of genetic parameters and breeding values for milk production characteristics and components in dairy goats is necessary, as these parameters are the indicators of the genetic progress that can be achieved when a good selection and mating program in this species has been implemented [3]. Knowledge of these parameters will help in the selection of superior individuals for these characteristics [4], allowing to increase the average performance of animals and in this way, establishing a constant expansion of the goat milk industry and its derivatives [5].

The obtaining of accurate estimates of heritability, variance components, and correlations is necessary to predict the expected selection response and predicted breeding values. Traits related to production, conformation, reproduction, production of milk, and its components are already considered in dairy cattle breeding programs in many countries [6]. Dairy goat breeds are characterized by a standardized uniform and stable selection criteria, which is always directed towards the common objective of identifying animals able to maximize production while minimizing resource costs.

Including environmental effects and genetic factors facilitates the design of more efficient and accurate models, whose prediction abilities improve the determination ability for future features and productive potential of resulting products. This not only helps to cover market demands earlier in the production chain but also at a lower cost, thus maximizing profitability. 

The use of multivariate techniques, such as categorical principal components and nonlinear canonical correlations [7], could be used to find the genetic (single-nucleotide polymorphisms, SNP, combinations) or environmental factors that explain the greatest variability for a certain trait. This provides a tool that enables building relationships that allow grouping animals according to their similar productive traits, basing on the underlying genetic correlations across the different traits considered. 

Once the groups of animals with desirable traits have been identified, these individuals constitute the basis for breeding programs aimed at improving traits such as productivity or fertility, among others [8,9,10]. Once superior individuals have been detected, identifying the genetic associations (dominance and additive effects) and the epistatic genetic relationships between traits of economic importance is essential as it may optimize the profitability of selection policies. In this context, multivariate analysis would allow us to address the decisions that may define the highest average-performing offspring in relation to the values of previous generations of heterosis and the increase in variability of the population [10,11].

For these reasons, the objective of this study was to evaluate and quantify the genetic repercussions of the inclusion of the additive and dominance components of SNP biomarkers clustered in principal component dimensions for the genes in the casein complex and of the epistatic relationships among such heritable units in predictive models on the estimation of genetic parameters (heritabilities and variance components), breeding values, and their accuracies for milk yield and components in Murciano-Granadina goats.

## 2. Materials and Methods 

### 2.1. Milk Yield Standardization and Composition Analysis

The productive management of Murciano-Granadina, as a polyestric breed is characterized by two annual kidding periods, with lactations lasting longer than 210-240 days [1]. Milk yield and components were estimated until 210 days of lactation and expressed in Kg [12]. Each goat’s real milk production (RP_j_) was computed through the equation described in Pizarro Inostroza et al. [7]. Official control policies are stated by the Royal Decree-Law 368/2005, on 8^th^ April 2005, regulating official milk yield controls for the genetic evaluation in the bovine, ovine, and caprine species of the Spanish Ministry of Agriculture (2005). Milking policies varied across farms depending on whether milking was performed every 4 or 6 weeks, during the morning or afternoon (AT4, AT4T, AT4M, A6, AT6M, and AT6T). First control (d_1_) and the last (d_2_) were assessed individually for every goat computing the days between the day the animal was born (BD) and the date of the first control (FC), and the days between the penultimate control (PC) and the last control (LC), respectively using the formulas in Pizarro Inostroza et al. [7]. To save interindividual differences that could be ascribed to differences in milking period among other factors, birthdate information, and the date on which several controls were performed until 210 lactation days were included to normalize milk yield for each goat. An average of approximately five controls was performed per goat. Standardized yield to 210 days per goat was calculated using the formula and model described in Pizarro Inostroza, et al. [7]. Milk sampling was carried out monthly and officially analyzed at the Milk Quality Laboratory, in Cordoba (Spain) to quantify protein, fat, solids, lactose content, and somatic cells count with a MilkoScan™ analyzer FT1.

### 2.2. Animals

Given the costs involved in genotyping, a selection process of goats that had been considered for milk yield standardization and composition analysis was implemented. This sample selection process aimed at genotyping a representative sample of animals for 48 SNPs in the casein complex from which complete records for several lactations existed. Hence, animals present in the herdbook of the National Association of Breeders of Goats of Murciano-Granadina breed (CAPRIGRAN) were ranked considering the most recent and updated official breeding values for milk yield and composition reported for all the animals published in 2015. Provided multiple traits are considered, we developed combined selection index (ICO) procedures following the premises in Van Vleck [13] to summarize the value of each individual comprising each of its partial values for milk yield and composition and these were computed for each animal using MatLab r2015a [14]. We decided not to include solids in the ICO, as redundancies may occur deriving from the relationship of this trait and fat or protein content. To determine the weights to apply to each trait, we considered the phenotypic relationship across milk yield and composition traits (except for solids), scoring their relevance as selection criteria when the breeding goal was milk yield and quality. In matrix notation, the weights to be applied on the selection index combining the partial scores of each modality were obtained as, $b=P^{-1}g$, where b is the vector of weights to be applied to each production or content trait, P is the phenotypic (co) variance matrix, and g is the vector of genetic (co)variances of every trait with each other. As a result, and considering the market demands, the weights for milk yield, fat, protein, and lactose followed the proportion of 1:1:1:1, respectively. The combined index used (ICO) was as follows:(1)$$ICO=\frac{PBV_{milkyield}W_{1}}{\mu_{milkyield}}+\frac{PBV_{fat}W_{2}}{\mu_{fat}}+\frac{PBV_{protein}W_{3}}{\mu_{protein}}+\frac{PBV_{lactose}W_{4}}{\mu_{lactose}},$$
where PBV is the predicted breeding value for each of the traits and animals included in the matrix; W_1_ is the weight for milk yield, W_2_ for fat, W_3_ for protein, and W_4_ for lactose in kg and standardized to 210 days; and μ the mean for each of the traits included in the ICO computed in Kg and at 210 days. After ICO was computed for each of the animals included in the matrix, we sorted a total of 200 animals from the whole routine milk recording of Murciano-Granadina goat breed in a ranking considering their ICO value obtained at the previous genetic evaluation. Animals with extreme PBVs may be less efficient and less balanced than we could expect at first. Furthermore, not all traits are affected to the same degree by selection for these extremes. For these reasons, initially, 200 animals were randomly selected and ranked as follows: 67 females presenting the lowest ICO values in the rank, 66 females with values around percentile 50, and 67 females presenting the highest ICO values in the rank, respectively. This sample selection process was performed to ensure that we worked on an adjusted representative sampling of the genotype distribution in the population. Out of these 200 animals initially considered, we discarded those whose phenotype registries were missing or incomplete. As a result, the final sample set for genotyping consisted of 2594 direct records of 159 studbook registered goats from which blood samples were taken for genotyping. Direct records were collected from 28 Southern Spanish farms in random periods, from 2005 to 2018. The age of the animals in the sample ranged from 1 to 9.15 years (1.57±1.11 years, mean ± sd). 

### 2.3. Genotyping

DNA isolation was performed through a modification of the procedure of Miller et al. [15]. We selected sixteen samples from Murciano-Granadina herdbook’s nonrelated individuals at random. Appendix A shows oligonucleotide sequences and SNPs promoters, UTRH3’ regions, and polymorphic exons. Polymorphic regions amplification was performed using Platinium High Fidelity (LifeTechnology, Carlsbad, CA, USA) PCR kit. MACROGEN sequencing service (Macrogen Inc., Seoul, Korea) was used to sequence the PCR product. MEGA7 software was used to analyze pherograms and Ensembl Genome Browser 97 database was used to evaluate polymorphic regions’ previous annotations for SNP information on markers assessed regarding minor and major allelic frequencies, location, among others [16]. We identified 48 SNPs in the individuals sampled. Genotyping was performed using the KASP assay (LGC Limited, Fordham, UK), analyzing raw allele calls with KlusterCaller software (LGC Limited, Fordham, UK). Heterozygosity values of around 40%, suggested the number of SNPs to be used as genomic controls was sufficient [17] to avoid effects derived from the stratification of the population.

### 2.4. Single-Nucleotide Polymorphisms (SNPs) Additive and Dominance Genetic Effects Identification and Codification and Dimensionality Reduction Using Linkage Disequilibrium (LD) and Categorical Principal Component Analysis (CATPCA)

Additive and dominance effects for each SNP were determined and encoded, as described in Pizarro Inostroza et al. [7]. Normal parametrization encoding was not used as it may potentially presume an ordinal relationship among the levels of each factor (every possibility within each SNPs in our case) with dominant homozygous being encoded as -1, heterozygous being encoded as 0, and recessive homozygous being encoded as 1 for additive genetic component and so on. Nonetheless, this assumption may be erroneous when the levels of these factors present in fact a nominal nature. Mistakenly presuming an ordinal distribution among the levels within a certain factor, which indeed is nominal, may condition sample properties and the interpretation of the outcomes derived from the analyses performed. 

Linkage disequilibrium (LD) and Categorical Principal Component Analysis (CATPCA) results were evaluated to select the minimum number of SNPs capturing the highest genetic variability of a given trait. The number of SNPs depends on the distribution of SNP allele frequencies and the existence of inter SNP LD. Minor allele frequency (MAF) enables differentiating between populations’ common and rare variants (MAF < 0.05). MAF was calculated using default settings for all SNPs by PLINK v1.90 [18]. Casein complex SNPs’ Linkage disequilibrium extent (LD) was calculated using HaploView software [19], scoring LD through D’ (normalized linkage disequilibrium coefficient) and r^2^ (linkage disequilibrium coefficient of determination). The total length of casein loci and distances between adjacent loci were determined following the premises presented by Dagnachew et al. [20]. Using a unique test to evaluate genotype-phenotype association, we lost the ability to control for confounding factors such as those derived from the structure of the population, genomic stratification, genetic environment, and gene interaction (epistatic relationships). Contextually, CATPCA can determine the variability explained by a certain set of factors that can derive from genome-wide analyses (single-nucleotide polymorphisms (SNP). This way, potential redundancies can be reduced through a rather, hence, more effective comparison space between SNPs [20,21]. This way, CATPCA, and particularly Bonferroni correction corrects for the bias derived from the inclusion of a large number of factors (increased likelihood of false positives), maximizing the variability explanatory power of factors combinations identifying and ruling out potential misinterpretations of SNP/phenotype associations [20,22]. Horne and Camp [23] proposed that principal component analysis (PCA) can evaluate SNP correlations determining clusters in LD (LD-clusters), setting an optimal set of group-tagging SNPs (gtSNPs) determining intra-genic diversity more efficiently and minimizing the necessary requirements for the evaluation of informative association. Unlike haplotype block (HB) and haplotype-tagging SNP (htSNP) methods based on Linkage Disequilibrium Analysis (LD), PCA technique (also applicable to CATPCA) does not require SNPs to be in Hardy-Weinberg’s equilibrium [24]. Furthermore LD-groups of SNPs do not need to be located in close DNA fragments. This way, there is a fraction of diversity variance, that of fragments located at different fragment being related, which would be lost and which can be computed through CATPCA [24,25]. Kaiser Varimax Rotation was used as well, as it corrects the bias derived from high correlations among factors and a small number of variables and zero correlations in the rest.

### 2.5. Study of OVERALs/Nonlinear Canonical Correlations (NLCC) to Identify and Encode Epistatic Effects

After identifying clustered dimensions from CATPCA analysis, the evaluation of the relationships established among such dimensions, using nonlinear canonical correlations (NLCC), can quantify the epistatic relationships that exist between SNPs that act as a single unit. NLCC can help identify similarities between SNP clusters. Despite several methods being available to infer the study of epistasis, such as standard linear canonical correlation (CCA) and Artificial Neural Network analysis (ANN), these normally fail in two aspects, namely the issues of local minima and overfitting [26]. The background supporting the methods to identify genetic interactions lies in the likelihood that the variability in the expression of a certain trait differs under the effects of such interaction (epistasis) [27,28]. Not only NLCC provide information on the relationship among SNPs clusters (identified by CATPCA), but also the degree at which they may affect productive traits variability. SNPs with component loadings of over |0.5| [29], were the most effective ones to identify relationships among SNP clusters because they were positioned farther from the mean [30]. 

Additionally, NLCC can be used to validate whether a presumed ordinal or nominal condition for the factors analyzed was in fact correctly assigned. Hence, if we consider the different possibilities (levels) within the same factor as nominal, either using normal parametrization, that is, encoding the dominant homozygous possibility as -1, the heterozygous possibility as 0, and dominant recessive possibility as 1 or alternatively encoding the dominant homozygous possibility as 1, the heterozygous possibility as 2 and the dominant recessive possibility as 3, the interrelationship across possibilities remains the same. Moreover, at the same time we prevented the problems arising from the use of negative numbers as a nominal category (level) occurring with many statistical programs. In this context, NLCC validation stems from the fact that this technique permits to classify factors into two or more sets and scale them as multiple nominal, single nominal, ordinal, or numerical. Then, the interpretation of their direction is obtained from the position of projected centroids which can be used as a validation for the encoding pattern used.

Explicitly, considering a variable as ordinal implies that the order of the levels (categories) within each factor must be preserved. Then, if actual and projected centroids are not separated, a presumable ordinal relationship does not exist, and ordinal variables should have been considered as nominal as in our study. 

The bidimensional solution for noncanonical linear correlation analysis explained 79.65% of the total variance in the phenotypic traits evaluated across SNP groups (57.56% of the variance for the first dimension and 42.43% for the second dimension). Out of the 48 SNPs evaluated, SNP18 was the most frequent (component loading >|0.5|, for dimension 2) and one of the most relevant when explaining intergroup epistatic variability for milk yield and components. 

### 2.6. Preliminary Statistical Assumption Testing

Milk yield and components have commonly been assumed to be normally distributed according to historical research. However, data obtained from the field often challenge researchers, as the methodology implemented or study conditions promote data to violate regular parametric assumptions. In this context, new alternatives for the assessment of such data offer new opportunities to fit the tools used to the characteristics of the data to be assessed [31]. Walsh’s Outlier non-parametric test was used to detect multiple outliers in the data set. Although this test requires a large sample size (n>220 for a significance level α of 0.05), it may be used whenever the data are not normally distributed. Data was purged and animals whose records fell outside the ranges reported for the breed in the bibliography were discarded. Parametric assumptions of normality, homoscedasticity, sphericity, and multicollinearity were tested on complete historical records in the official dairy control of the Murciano-Granadina breed up to 2018 (n = 2,359,479 records of 151,997 goats) for milk yield, content (fat, protein, solids, and lactose) and somatic cells count. The Shapiro-Wilks Francia normality routine of the test and distribution graphics package of the Stata Version 15.0 software test was used to test the normality (Appendix A). The rest of the parametric assumptions (Levene’s test to evaluate homoscedasticity and Mauchly’s W test to evaluate sphericity and Tolerance and Variance Inflation Factor to test for multicollinearity, respectively) were performed using SPSS Statistics for Windows statistical software, Version 25.0. 

Parametric assumptions can be violated as a result of distribution irregularities or bias occurring during the selection process of research samples. However, despite parametric assumptions could have been violated by Murciano-Granadina historical records of milk yield and composition, data for milk performance and composition from the goats at the same lactational stage has been reported to presumably follow a normal distribution [7,12,32], which may support the implementation of parametric approaches. Hence, parametric assumptions were tested again, clustering the records from goats at the same lactation, to detect potentially occurring biases in the distribution of our data, as a way to reinforce our decision on whether to implement parametric or nonparametric approaches. 

### 2.7. Non-Genetic and Genetic Fixed Effect Statistical Analysis 

As stated by Bidanel [33], increasingly sophisticated genetic evaluation models may undoubtedly contribute to increasing the efficiency of animal breeding plans. In these regards, the same author suggested the need to adequately describe the structure of dispersion parameters, hence, the need to consider realistic genetic models. However, careful model checking and validation is a necessary prior step to ensure that the proposed model is fully justified. Additionally, Andonov et al. [34] reported the likelihood-ratio test, Akaike information criterion, and mean-squared error of prediction favor more complex models. In this context, models fitting animals’ additive genetic merit as the only genetic effect, enable the approximation of second derivatives of the likelihood function to provide appropriate estimates of sampling covariances between estimates. However, models containing other genetic effects, either these are non-additive genetic or maternal genetic effects, often report large negative sampling correlations between estimates, which results in a shape of the likelihood surface which in general does not allow second derivatives to be approximated by numerical differentiation [35]. Thus, particular consideration must be provided to the statistical selection process of fixed effects.

Following these premises and the preliminary analyses performed, parametric assumptions were also tested on our field data. As preliminary tests and our study data had violated parametric assumptions, a nonparametric approach was suggested. The Kruskal–Wallis H test was performed to identify differences in the distribution of the data for milk yield (Kg), fat, protein, solids, lactose content (%), and somatic cell count (cs/mL) across the levels of the same factor (genetic factors whether they are additive or not (dominance and epistasis) and nongenetic factors). Out of all possible level comparison pairs, only statistically significant pairwise comparisons were considered by Dunn Test. Then, Bonferroni correction was applied to reduce the likelihood of an increased Type I error potentially deriving from redundancies resulting from the inclusion of an excessive number of factors considering the relatively limited sample of our study. Then, partial eta square (ηp^2^) was computed as it expresses inter category differences and their associated error variance as a percentage. Afterwards, univariate tests must be carried to isolate the pairs of categories of each factor between which a significant difference in the mean value for the independent variable exist. Past research suggests ηp^2^ may be more appropriate than eta square (η^2^) after performing a multifactorial design, as ηp^2^ provides a score for the association strength between independent factors and dependent variables, excluding the variance produced by the rest of factors considered in the model as suggested by Brown [36].

When parametric assumptions are not fulfilled, the Kruskal–Wallis test is an alternative to MANOVA (Multivariate ANOVA). Kruskal–Wallis H statistic is based upon a single independent factor accounting for the variance explained of a dependent variable with no additional factor contributing to the explanation of such variance at the same time, a particular situation which makes ηp^2^ be equal to η^2^. Contextually, this may be of remarkable importance when we consider non-possibly overlapping empirical variables as suggested in Pizarro et al. [32]. A complete description of the statistical analyses carried out on the nongenetic and genetic fixed effects included in the genetic model can be found in Pizarro et al. [32].

The analysis of the relationship between factors such as Days in Milk (DIM), Days to first control (DFPC), and Days from last control to drying (DLD), number of kids born alive and number of kids born dead and milk yield (kg) and component variables was performed using Pearson product-moment correlation coefficient. Kruskal–Wallis H test, Dunn test, and Bonferroni’s correction and Pearson Product-Moment correlation analysis were carried out with SPSS Statistics software for Windows, version 25.0 was used to perform statistical tests.

### 2.8. Genetic Model Comparison, Phenotypic and Genetic Parameter Estimation

Genetic analyses were performed using the data provided by the Murciano-Granadina Breeders Association, derived from the actions implemented within the scope of its breeding program. Overall, the pedigree included records of 244,046 animals with indirect or direct records related through at least one known ancestor (232,804 does and 11,242 bucks), while the phenotype field database comprised 2594 direct records of 159 goats evaluated from 2005 to 2018 that have been genotyped. Therefore, a multitrait animal mixed model with repeated measures was used to estimate (co) variance components, and heritability, repeatability, phenotypic and genetic correlations and standard errors of such correlations. In matrix notation, the following multitrait animal model with repeated measures was used:(2)$$Y_{\mathrm{yfpdmlsc}}=\mu +{\mathrm{Far}}_{a}+{\mathrm{Pye}}_{b}+{\mathrm{Pmon}}_{c}+{\mathrm{Pse}}_{d}+{\mathrm{Bnum}}_{e}+{\mathrm{Cye}}_{f}+{\mathrm{Cmon}}_{g}+{\mathrm{Cse}}_{h}+{\mathrm{Nc}}_{i}+{\mathrm{Mrout}}_{j}+\;{\mathrm{Bty}}_{k}+{\mathrm{Dye}}_{l}+{\mathrm{Dmon}}_{m}+{\mathrm{Dse}}_{n}+{\mathrm{DIM}}_{o}+\mathrm{DFPC}+{\mathrm{DLD}}_{q}+{\mathrm{Aln}}_{r}+{\mathrm{Den}}_{s}+\mathrm{PC}1_{t}+\mathrm{PC}2_{u}+\mathrm{PC}3_{v}\;+\mathrm{PC}4_{w}+\mathrm{PC}5_{x}+\mathrm{PC}6_{y}+\mathrm{PC}7_{z}+{\mathrm{NLCC}}_{\mathrm{aa}}+b_{1}A_{\mathrm{ab}}+b_{2}^{2}A_{\mathrm{ab}}+{\mathrm{Animal}}_{\mathrm{ac}}+{\mathrm{PE}}_{\mathrm{ad}}+\varepsilon_{\mathrm{yfpdmlsc}},$$
where Y_yfpdmlsc_ is the separate score of milk yield (y) and components (fat (f), protein (p), solids (dm), lactose (l) in kg and somatic cells (sc) count cs/mL) for a given animal; μ is the overall mean; Far_a_ is the fixed effect of the ath farm/owner (a = 28 farms); Pye_b_ is the fixed effect of the bth year of parturition (b = 2005–2018); Pmon_c_ is the fixed effect of the cth month of parturition (c = January to December); Pse_d_ is the fixed effect of the dth season of evaluation (d = Autumn, Winter, Summer, and Spring); Bnum_e_ is the fixed effect of the eth birth number (e = 1-9 Birth); Cye_f_ is the fixed effect of the fth control year (f = 2005–2018); Cmon_g_ is the fixed effect of the gth control month (g = January to December); Cse_h_ is the fixed effect of the hth control season (h=Autumn, Winter, Summer, and Spring); Nci is the fixed effect of the ith number of control (i=1-31 controls); Mrout_j_ is the fixed effect of the jth milking routine (j = A4, AT4T, AT4M, A6, AT6M, AT6T); Bty_k_ is the fixed effect of the kth birth type (k = Simple, Double, Triple, abortion in lactation); Dye_l_ is the fixed effect of the lth drying year (l = 2005–2018); Dmon_m_ is the fixed effect of the mth drying month (m = January to December); Dse_n_ is the fixed effect of the nth drying season (n = Autumn, Winter, Summer, and Spring); DIM_o_ is the fixed effect of the oth number of days in milk (o = 1-183 days); DFPC is the fixed effect of the pth number of days to first control (p = 1-226 days); DLD_q_ is the fixed effect of the qth number of days from last control to drying (q = 1-61 days); Aln_r_ is the fixed effect of the rth number of kids born alive (r = 0-5 kids); Den_s_ is the fixed effect of the sth number of kids born death ( s= 0-3 kids); PC1_t_ is the fixed effect of the tth additive and dominance effect of SNPs in PC1 (t = 1-17); PC2_u_ is the fixed effect of the uth additive and dominance effect of SNPs in PC2 (u = 1-6); PC3_v_ is the fixed effect of the vth additive and dominance effect of SNPs in PC3 (v = 1-17); PC4_w_ is the fixed effect of the wth additive and dominance effect of SNPs in PC4 (w = 1-20); PC5_x_ is the fixed effect of the xth additive and dominance effect of SNPs in PC5(x = 1-15); PC6_y_ is the fixed effect of the yth additive and dominance effect of SNPs in PC6 (y = 1-14); PC7_z_ is the fixed effect of the zth additive and dominance effect in PC7 (z = 1-12); NLCC_aa_ is the fixed effect of the aath epistatic nonlinear canonical correlation between SNPs (NLCC) (aa = 1-10); age in months was considered a linear and quadratic covariate, hence $b_{1}$ and $b_{2}^{2}$ are the linear and quadratic regression coefficients on the age of evaluation (A_ab_), Animal_ac_ is the random additive genetic effect of the acth goat, PE_ad_ is its permanent environmental effect of each goat, and ε_yfpdmlsc_ is the random residual effect. 

Afterward, genetic factors (additive and non-additive, dominance and epistasis) were then excluded from the model to isolate their potential effects on the traits measured. The rest of the terms included in the model were the same as those defined above. In matrix notation, the multitrait animal model with repeated measures excluding genetic factors was as follows: (3)$$Y_{\mathrm{yfpdmlsc}}=\mu +{\mathrm{Far}}_{a}+{\mathrm{Pye}}_{b}+{\mathrm{Pmon}}_{c}+{\mathrm{Pse}}_{d}+{\mathrm{Bnum}}_{e}+{\mathrm{Cye}}_{f}+{\mathrm{Cmon}}_{g}+{\mathrm{Cse}}_{h}+{\mathrm{Nc}}_{i}+{\mathrm{Mrout}}_{j}\;+{\mathrm{Bty}}_{k}+{\mathrm{Dye}}_{l}+{\mathrm{Dmon}}_{m}+{\mathrm{Dse}}_{n}+{\mathrm{DIM}}_{o}+{\mathrm{DFC}}_{p}+{\mathrm{DLD}}_{q}+{\mathrm{Aln}}_{r}+{\mathrm{Den}}_{s}+b_{1}A_{t}+b_{2}^{2}A_{t}+\;{\mathrm{Animal}}_{u}+{\mathrm{PE}}_{v}+\varepsilon_{\mathrm{yfpdmlsc}},$$

MTDFREML software package [34] was used to carry out restricted maximum likelihood-based univariate analyses to calculate estimates for heritabilities and variance components. Afterward, bivariate analyses to estimate covariates and genetic and phenotypic correlations. The iteration process used sought a convergence criterion level of 10^−12^. Link functions and their mathematical development is shown in Boldman et al. [37]

### 2.9. Non-Genetic Best Linear Unbiased Estimators (BLUE) for Fixed Effects and Covariates and Best Linear Unbiased Predictors/Breeding Value Prediction (BLUP, PBVs) 

After reaching convergence, best linear unbiased estimators for non-genetic fixed effects and covariates (BLUE) and best linear unbiased predictors for random effects, i.e., predicted breeding values (BLUP, PBVs), were directly estimated with MTDFREML software [34]. Accuracies and reliabilities for milk yield, fat, protein, and solids for each animal in the matrix using the same software.

### 2.10. Predicted Breeding Values (PBV), Standard Error of Prediction (SEP), Accuracies (RTi), and Reliability (Rap) Comparison

The genetic evaluation performed in the present study was undertaken using BLUP, restricted maximum likelihood methodology and a deep pedigree of 244,046 animals that ranged from zero to six complete generations or zero to fifteen maximum generations in length (depending on the individual). Pearson product-moment correlation analysis between the predicted breeding values (PBV) for milk yield, protein, fat, and solids (expressed in kg) obtained using both the model including the αS1-casein genotype and the one excluding it for all the animals included in the pedigree (n = 244,046) of Murciano-Granadina breed goats was carried out to check for the replicability of the results across models.

### 2.11. Ethics Approval and Consent to Participate 

The present study works with records provided by the National Association of Breeders of Murciano-Granadina breed Goats (CAPRIGRAN) rather than live animals directly, hence no special permission was compulsory.

## 3. Results

### 3.1. SNPs Dimensionality Reduction Using Linkage Disequilibrium and CATPCA 

Once the additive and dominance effects have been encoded for each of the 48 SNPs, LD and Categorical Principal Components (CATPC) were studied to select which SNPs may capture the highest genetic variability in milk yield and its components. As suggested by Pizarro Inostroza et al. [7], we can infer that CATPCA method overcomes existing htSNP methods, as it proposes the optimal number of SNPs to choose while it simultaneously maximizes the amount of genetic variation explained by a candidate gene or a group of candidate genes, such as the casein complex in our study, using a minimal number of SNPs [23].

Seven dimensions were identified at a Cronbach’s alpha level of over 0.803 (as described in Pizarro Inostroza et al. [7]), which is over the minimum level to support reasonable internal consistency of the elements (SNPs) selected, hence, the reliability is acceptable after redundant SNPs are removed. These clustering dimensions were encoded and included in the genetic model to quantify for additive and dominance effects. Dagnachew and Ådnøy [38] reported the occurrence of high component loadings in the same categorical principal component dimension to be a sign of higher correlations among the SNPs loading over |0.5| for that particular dimension or what is the same of LD-groups of SNPs. 

Out of the 48 SNPs studied, a total of 40 SNPs contributed to the seven-dimensional model in a meaningful way (factor loadings>|0.5| for CATPCA), then the different components (PC1, PC2, PC3, PC4, PC5, PC6, and PC7) were best described by the SNPs highlighted included in the red rectangle in Figure 1. SNP7, 9, 11, 21, 27, 30, 33, and were discarded given they did not participate in any of the dimensions identified (confounding or variance explaining redundant SNPs). 

### 3.2. Study of OVERALs/Nonlinear Canonical Linear Correlations (NLCC) to Identify and Encode Epistatic Effects

The analysis of nonlinear canonical correlations identified two highly loaded dimensions (0.917 and 0.676 eigenvalues for dimensions 1 and 2, respectively). Average loss (2-1.593 = 0.407). Hence, a bidimensional solution was chosen, so 1.593/2 = 79.65% of the variation was explained by the SNPs highly loading (component loading>|0.5|) for both dimensions. 0.917/1.593 = 57.56% of the actual fit was calculated by the first dimension and 0.676/1.593 = 42.43% by the second dimension was not high. Appendix A shows a summary of the results of the epistatic relationship through NLCC, SNPs explaining intergroup variability and reinforcing epistatic interaction (multiple fit>0.1), alleles for each SNP, their relative frequency and dominance ratios. 

### 3.3. Non-Genetic and Genetic Fixed Effect Analysis 

Appendix A presents a summary of the results for the Kruskal–Wallis H test and partial eta squared coefficient (ηp^2^), and Pearson Product-Moment Correlation Coefficient (ρ) providing information regarding the existence of differences in the distribution across levels within the same factor and the variance explanatory power of these independent variables factors.

### 3.4. Genetic Model Comparison, Phenotypic and Genetic Parameters Estimation

The estimates for heritability, genetic, phenotypic, and permanent environmental variance obtained through restricted maximum likelihood methods for models including and excluding the additive and dominance effects for SNPs in Principal Components from 1 to 7 and the epistatic nonlinear canonical correlation between SNPs as fixed effects are shown in Table 1. The estimated results for the production of milk, fat, protein, solids, lactose (%), and somatic cells (cs/mL) when additive, dominance, and epistatic interactions of the SNPs of the genes in the casein complex (*CSN1S1*, *CSN2S2*, *CSN2*, *CSN3*) considered as a fixed effect were 0.46; 0.22; 0.25; 0.21; 0.30, and 0.43, respectively. Somehow, comparatively, when these factors were excluded, heritabilities were, 0.21; 0.24; 0.24; 0.24; 0.22, and 0.20, respectively (Table 1). Genetic (r_G_) and phenotypic (r_P_) correlations [39] are shown in Table 2.

### 3.5. Predicted Breeding Values (PBV), Standard Error of Prediction (SEP), Accuracies (RTi), and Reliability (Rap) Comparison

The comparison of predicted breeding values did not report significant differences in regard to whether genetic effects (additive, dominant or epistatic effects) were included or not. Contrastingly, Table 3 reports descriptive statistics for standard error of prediction (SEP) and accuracies (RTi) between models when genetic effects (additive, dominance or epistatic effects) were included or excluded. Simultaneously, Table 4 shows significant correlations were found when predicted breeding values were compared between the model which considered genetic effects (additive, dominance, and epistatic relationship) and the one which did not, for milk yield, composition traits, and somatic cells counts.

## 4. Discussion

Genomic selection could be basically considered a form of marker-assisted selection in which a very large number of genetic markers (10.000 up to ≈800.000 SNPs) are used. Quantitative trait loci (QTL) are sections of DNA that correlate with the variation of a quantitative trait in the phenotype of a population. All QTL are closely linked at the chromosomes with at least one marker. Under these premises, conclusions drawn from high scale genomic selection may be applicable to marker-assisted selection using lower numbers of SNPs. The high number of genetic markers used in genomic selection can be used as input in a genomic formula that predicts the breeding value of an animal, so does the lower set of SNPs used in marker-assisted selection. Statistically, this reduction in the number of SNPs translates into a reduction in the values of adjusted determination coefficients (adjusted R^2^), thus, variance explanatory power, when compared to those models, comprises higher numbers of SNPs used in genomic selection. This is supported by Pizarro et al. [32], who reported determination coefficients or percentages of variance explained that ranged from only around 15% to 40% for the models comprising the 40 SNPs in our study.

Estimates of genetic parameters, among them heritability and correlations, are of great importance, as they provide us with the necessary information that allows us to select the best animals according to our productive needs [1]. This is the reason why Murciano-Granadina breeding programs have focused on increasing several economically important traits for greater productivity. The application of inappropriate multivariate statistical techniques may alter animal hierarchization or ranking depending on breeding value when milk yield and components are studied, reporting erroneous information can result in detrimental effects for the programs implemented. This addresses the need to design analysis plans fitting the specific situation occurring at a certain population [40].

In this context, principal components analysis (PCA) reduces the number of originally correlated variables into a smaller set of uncorrelated variables, maintaining most of the original variability and reducing dimensionality to a new set of variables, under the assumption of losing as little information as possible, which improves the descriptive performance of models [41]. The PCA technique has been successfully incorporated into genetic assessments in horses [42], cattle [43], dairy [44], and for the analysis of reproductive traits in several bovine breeds [45,46]. Principal component methods have commonly been aimed at disentangling the potential redundancies among traits measured rather than directly implemented towards the identification of a common explicative structure underneath genetic effects (whether it is additive, dominance, or epistatic), there have been some attempts to apply such methods in goats’ genetic evaluations [47]. Unfortunately, clustering based on principal components analysis, performed with SNP did not reveal any major distinct groups, hence, its application is not feasible.

Benradi et al. [48] reported heritabilities for the Murciano-Granadina breed, which were slightly lower than those found in this study for milk yield and fat content (0.18, 0.16 with single-trait analysis), while values for protein content matched those in our study (0.25). Pizarro et al. [32] reported superior heritability for protein content of 0.53 in the Murciano-Granadina breed, when genotype for αS1-casein was included, which may suggest the model used a rather complex diversification of genetic effects into additive, dominance, and epistatic effects may improve the performance of the model used, which may base on an improved control of such genetic relationship within and between the genes in the casein complex. 

In regards to somatic cells count (cs/mL), lower results were reported by recent studies, suggesting heritabilities may range from 0.20 to 0.24 in Saanen and Alpine goats [48,49,50,51]. However, these studies did not involve any genetic factor but nongenetic ones in the model, thus results are not comparable. These heritability values enable the potential selection of animals considering the levels of somatic cells as a selection criterion. Somatic cell counts must be considered as a potential trait for selection given the implications that the migration of neutrophils from blood to mammary gland as a response to infection has on the quality, hence, the value of milk [49,52,53,54]. Higher heritabilities in our results when genetic effects are included in the model compared to those obtained when these effects are excluded suggest including genetic effects may improve selection possibilities, which may derive from a better performance of the models used.

Moderate heritability values for lactose content were reported. These values are in agreement with the heritability values of 0.27 for lactose content reported in the literature for Polish White Improved (PWI) and Polish Fawn Improved (PFI) does [48]. Contextually, heritability estimates confirm the existence of a considerable level of genetic diversity that allows the possibility of continuing to improve the production of milk, fat, protein, solids, lactose, and somatic cell content in the breeding programs and selection of goats of Murciano-Granadina breed, even if they are highly selected breeds [49,55].

Heritability standard errors for milk yield and fat content, protein, solids, lactose, and somatic cells including SNPs dominance and additive effects and epistatic interactions among casein genes (*CSN1S1, CSN2S2, CSN2, CSN3*) as fixed effects (0.05; 0.01; 0.01; 0.05; 0.05, 0.01; 0.07 for milk yield, fat, protein, solids, lactose, and somatic cell counts, respectively), suggested the outcomes from the models tested report reliable enough values. When such genetic effects were not considered in the model the values for standard error of prediction were 0.01; 0.01; 0.02; 0.01, 0.01; 0.05 for milk yield, fat, protein, solids, lactose, and somatic cell counts, respectively (Table 1), which remained within the ranges of estimated values in other dairy goat populations [55,56,57], indicating that the estimation of the parameters studied in our samples is comparable.

Genetic correlations, including additive effects, dominance effects, and epistatic interactions among SNPs of the genes of the casein complex were negative between milk yield and protein content (−0.42), milk and lactose performance (−0.07), lactose and protein content (−0.20), and lactose content and somatic cells (−0.22). When SNPs genetic effects were excluded from the model genetic correlations between milk yield, fat, protein, solids contents and somatic cell count (−0.29; −0.33; −0.34 and −0.18, respectively), fat and lactose content (−0.18), protein and lactose (−0.16), and lactose and somatic cells (−0.32). Very similar values were found in the Saanen and Alpine breeds [3].

These negative values have been reported for genomic correlations between loci and have been attributed to the alteration of the relationship between traits under directed selection [58], as it would happen in our study. However, low somatic cell counts are related to a lower prevalence of intramammary infection [59]. Hence, these favorable genetic correlations may support the use of a selection rate that will adequately weigh these traits to maximize the economic response for farmers. Phenotypic correlations estimated from the models including and excluding SNPs additive, dominance and epistatic interactions among casein genes were similar. Furthermore, very similar values were found in Saanen breeds; Alpina and Toggenburg [3,60].

Negative genetic and phenotypic correlations between milk yield and fat content, protein solids, lactose, and somatic cells revealed unfavorable associations, which could negatively affect the quality of dairy products, particularly those of special relevance for the cheese industry. Therefore, dairy goats should be selected using indexes that consider the phenotypic relationships between these traits but also include SNPs epistatic interactions (as a rather diversified element of quantification of inter and intralocus gene relationships). Including the existing association of such traits with yield and a firmer cheese curd [61] can maximize the economic response of the markets that commercialize goat’s milk products.

The highest genetic correlations were found for fat and solids contents (0.85), protein and solids contents (0.60), and fat and protein (0.43) when genetic effects (additive, dominance, and epistatic effects) were considered in the model. Contrastingly, when such genetic effects were excluded, the genetic correlation between the same traits reduced to 0.73; 0.51 0.39, respectively. In this context, genetic correlations may be considered to improve the profitability of the outputs of selection given the implicit potential increase in the content of components that are intrinsically related to cheese production, as has been reported for other high-quality cheese production dairy goat breeds [62,63]. Phenotypic correlations are in the range for other dairy goat populations observed in the literature. The highest correlations were reported for fat and solids content, 0,96 and 0,95, when genetic effects were included and excluded from the model, respectively [64,65].

It should be noted that there is a moderate similarity between the values of genetic correlations in both models since we evaluated the same traits in the same population set. The positive correlations found for fat and other components such as protein or solids had been previously described by Verdier-Metz et al. [66] who assessed cheese yield of milk by calculating it through fresh yield by dividing the weight of the fresh curd by the amount of milk used for cheese and solids production by multiplying the fresh yield by the value of the dried matter of the molded curd. These authors reported a wide range of values (55 to 85 g/kg) for the ratio of fats and proteins across different types of manufactured milk. The same proportion linearly accounted for 77% of fresh yield and 87% of solids yield variability. This indicates that the values reported in this study in relation to fat/protein/solids could produce an increase in the economic value of the Murciano-Granadina goat, which may contribute to a better yield of cheese production [62].

The relationship between fat and protein is not only important from a nutritive point of view, but also from a cheese performance-enhancing perspective. Milk component enrichment must be balanced as increases in protein that are not simultaneous to fat increases, will result in very aqueous and less fatty cheeses, which may not fulfill the needs of the markets [67].

The use of functional models provides an alternative vision that reflects the physiological or molecular interactions that more precisely describe the gene mapping of the economically important phenotypes involved in productive traits such as milk or cheese production [68]. A substantial body of theory and simulation has shown that genetic interactions can be an important determinant of heritable variation and therefore the response to selection of populations [69,70].

Despite many empirical studies that demonstrate genetic interactions, key empirical evidence is scarce on how the epistasis influences heritability, affects functional traits, or can be controlled and modeled. As a common factor, almost all the studies found in literature lack a clear population context which is crucial given heritability is a specific parameter, not only of the population being measured but, the tools used to measure it and the model implemented.

The lack of significant differences in regard to whether genetic effects (additive, dominant, or epistatic effects) were included or not may suggest the prediction of breeding values for each individual may be similar across models. Significant correlations were higher for standard error of prediction of milk yield, fat, and protein content predicted breeding values. However, despite being significant, these values drastically reduced when the results for reliability and accuracy of predicted breeding values were compared between the two models, including and excluding genetic effects. This may suggest the inclusion of genetic effects as fixed effects may increase the estimative power and accuracy of the model used to perform genetic evaluations for economically important traits linked to milk yield and its components.

These findings are in agreement with those in literature according to which when a relatively low number of genotyped animals is assessed through a shallow pedigree, that is, individuals which are likely to be distantly related or unrelated are contrasted within a considerably large pedigree and all breeding values estimated using BLUP are zero. Conclusively, BLUP may still predict a breeding value with a significant accuracy when animals in the test population (genotyped animals in our field database) and reference population (whole pedigree) may share a distant relationship and accuracy reduced to close to zero when animals in the pedigree and study sets are poorly related, distantly related or unrelated [71].

## 5. Conclusions

The heritabilities and correlations between fat, protein, and solids content must be considered given their implication in the nutritional and commercial quality of several dairy products. Additionally, lactose and somatic cells count genetic parameters may also be critical traits to consider given their implications with milk’s health quality. These intertrait relationships may address which selective paths may be more profitable, hence, they should be considered when targeting the maximization of the commercialization potential of goat milk as a raw material for cheesemaking industry. Indexes comprising the evaluation of these traits considering additive, dominance, and epistatic relationships among casein complex genes or SNPs could lead to remarkable increases in the outcomes derived from selective practices and indirectly in the economic value of dairy breeds. As suggested by our results, the inclusion of casein complex SNPs additive, dominance, and epistatic effects in the model used in genetic evaluations for milk yield and components may increase models’ estimative power and the accuracy of the breeding values obtained after dairy goat genetic evaluations when compared to models lacking such effects. This may translate into the enhancement of the genetic progress of the breed and strengthen the international competitiveness of Murciano-Granadina breed in the dairy goat industry.

## Figures and Tables

**Figure 1 genes-11-00309-f001:**
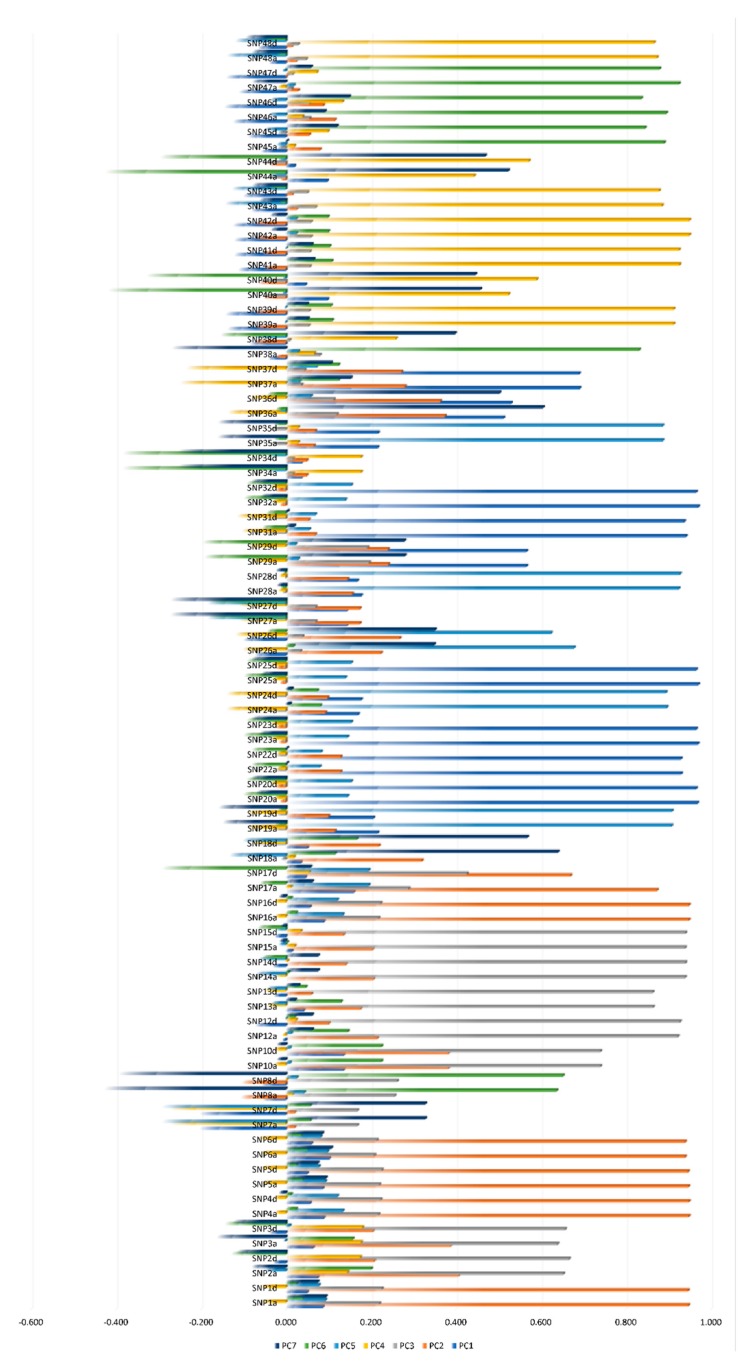
Categorical principal component analysis (CATPCA). Rotated component loadings for each dimension included in the rotated model using the Varimax method with Kaiser normalization (Red rectangle marks all single-nucleotide polymorphisms (SNPs) significantly loading (≥|0.5|) across the seven PC dimensions, hence, contributing to the explained variance). Accessed from Pizarro Inostroza et al. [7].

**Table 1 genes-11-00309-t001:** Estimated components of variance, heritability (h^2^), and standard error (SE) for milk yield (kg), protein (kg), fat (kg), solids (kg), lactose (kg), and cells somatic (cs/mL) obtained from multivariate analyses through REML methods in goat milk including and excluding αS1-Casein, αS2-Casein, β-casein, and κ-casein additive, dominance, epistatic factors as a fixed effect.

Model/Genetic Effects as a Fixed Effect	Trait (Kg)	$\sigma_{a}^{2}$	$\sigma_{p}^{2}$	$\sigma_{pe}^{2}$	$\sigma_{e}^{2}$	h^2^±SE
Including genetic effects as a fixed effect.	Milk yield	0.75450	1.63632	0.140011	0.74180	0.46±0.05
	Fat	0.37663	1.72151	0.204217	1.14066	0.22±0.01
	Protein	0.06216	0.24909	0.0276599	0.15927	0.25±0.01
	Solids	0.53164	2.58194	0.285996	1.76430	0.21±0.05
	Lactose	0.03361	0.11198	0.0213750	0.05699	0.30±0.01
	Somatic cells	1,450,503.8674	3,373,095.25	36,251.9	1,886,339.4833	0.43±0.07
Excluding genetic effects as a fixed effect.	Milk yield	0.34930	1.65896	0.172186	1.13747	0.21±0.01
	Fat	0.42176	1.72965	0.177331	1.13056	0.24±0.01
	Protein	0.06541	0.26935	0.0286661	0.17527	0.24±0.02
	Solids	0.67591	2.80513	0.291377	1.83785	0.24±0.01
	Lactose	0.02505	0.11168	0.0114023	0.07523	0.22±0.01
	Somatic cells	483,509.3208	2,368,170.87	244,535.	1,640,126.5503	0.20±0.05

**Table 2 genes-11-00309-t002:** Estimated heritabilities (h^2^) (diagonal), phenotypic (r_P_) (above diagonal), and genetic (r_G_) (below diagonal) correlations for milk yield (kg), protein (kg), fat (kg), solids (kg), lactose (kg), and somatic cells (cs/mL) obtained in bivariate analyses through REML methods in goat milk including and excluding αS1-Casein, αS2-Casein, β-casein, and κ-casein additive, dominance, epistatic factors as a fixed effect.

Model/Genotype as a Fixed Effect	Trait	Milk Yield	Fat	Protein	Solids	Lactose	Somatic Cells Count
Including genotype as a fixed effect	Milk yield	0.46	−0.41	−0.48	−0.46	0.12	−0.24
	Fat	−0.42	0.22	0.56	0.96	−0.10	0.16
	Protein	0.09	0.43	0.25	0.71	−0.29	0.29
	Dry mater	0.08	0.85	0.60	0.21	−0.03	0.15
	Lactose	−0.07	0.05	−0.20	0.03	0.30	−0.38
	Cells somatic	−0.27	0.18	0.18	0.16	−0.22	0.43
Excluding genotype as a fixed effect	Milk yield	0.21	−0.33	−0.41	−0.38	0.09	−0.25
	Fat	−0.29	0.24	0.46	0.95	−0.10	0.16
	Protein	−0.33	0.39	0.24	0.66	−0.28	0.28
	Dry mater	−0.34	0.73	0.51	0.24	0.03	0.14
	Lactose	0.08	−0.08	−0.16	0.02	0.22	−0.36
	Cells somatic	−0.18	0.13	0.24	0.12	−0.32	0.20

^a^ h^2^ ± SE; ^b^ r_P_ ± SE_;_
^c^r_G_ ± SE.

**Table 3 genes-11-00309-t003:** Summary of the descriptive statistics for standard error of prediction (SEP) and accuracies (RTi) between models when genetic effects (additive, dominance, or epistatic effects) were included or excluded.

Model		Including Additive, Dominance and Epistatic Genetic Effects				Excluding Additive, Dominance and Epistatic Genetic Effects			
	Descriptive	Min	Max	Mean	SD	Min	Max	Mean	SD
Parameters									
Milk yield (Kg)	SEP	0.59	0.69	0.59	0.01	0.87	1.02	0.87	0.01
	Rti	0.00	0.66	0.01	0.03	0.00	0.02	0.00	0.00
Fat (Kg)	SEP	0.00	0.76	0.65	0.01	0.61	0.72	0.61	0.01
	Rti	0.00	0.96	0.01	0.05	0.00	0.02	0.00	0.00
Protein (Kg)	SEP	0.17	0.30	0.26	0.00	0.25	0.29	0.25	0.00
	Rti	0.00	0.76	0.01	0.04	0.00	0.02	0.00	0.00
Solids (Kg)	SEP	0.00	0.96	0.82	0.02	0.73	0.85	0.73	0.01
	Rti	0.00	0.96	0.02	0.06	0.00	0.02	0.00	0.00
Lactose (Kg)	SEP	0.13	0.19	0.16	0.00	0.18	0.21	0.18	0.00
	Rti	0.00	0.61	0.01	0.03	0.00	0.02	0.00	0.00
Somatic cells count (cs/mL)	SEP	554.46	815.37	696.41	8.81	1204.37	1412.25	1206.80	14.82
	Rti	0.00	0.60	0.01	0.03	0.00	0.02	0.00	0.00

**Table 4 genes-11-00309-t004:** Pearson Product Moment (ρ) correlation comparison of Predicted breeding values’ (PBVs), Standard error of prediction (SEP) and accuracies (RTi) between models when genetic effects (additive, dominant, or epistatic effects) were included or excluded.

PBV Parameters	Pearson Product Moment Correlation
SEP Milk yield (Kg)	0.994 **
RTi Milk yield (Kg)	0.103 **
SEP Fat (Kg)	0.674 **
RTi Fat (Kg)	0.097 **
SEP Protein (Kg)	0.671 **
RTi Protein (Kg)	0.099 **
SEP Solids (Kg)	0.022 **
RTi Solids (Kg)	−0.009
SEP Lactose (Kg)	0.045 **
RTi Lactose (Kg)	0.012 *
SEP Somatic cells (cs/mL)	0.036 **
RTi Somatic cells count (cs/mL)	−0.010

SEP: Standard error of prediction; RTi: accuracy. *p* < 0.01 *; *p* < 0.05 **.

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
