# Peer review of "Integrating Casein Complex SNPs Additive, Dominance and Epistatic Effects on Genetic Parameters and Breeding Values Estimation for Murciano-Granadina Goat Milk Yield and Components"

_genes, 2020, doi:10.3390/genes11030309_

Round 1

Reviewer 1 Report

The resubmitted paper was reconstructed, and some parts were additionally described. The questions asked to the primary submission have been answered and clarified my doubts, so I'm not raising any more objections to the work (from the methodological perspective). Moreover, it is gratifying to see that since 2005 (Martinez et al., 2010) the breed is under sound scientific attention and aspects like the Bulmer effect were recognized, and selection (in general) is in good hands.

Therefore, I recommend accepting a paper. However, due to the hard the reviewer Word mode reading, I would like to see the manuscript without adjudication. In an example, the quality of Figure 1 is unreadable in reviewer Word mode or/and should be improved. I also missed different font formatting in Supplementary Tables (i.e. Table 3).

Reviewer 2 Report

The discussion of outliers is still not adequately address (Lines 357-358).